# LEARNING TO LEARN WITHOUT LABELS

Luke Metz, Niru Maheswaranathan, Brian Cheung, Jascha Sohl-Dickstein

Google Brain, Berkeley
{lmetz, nirum, jaschasd}@google.com, bcheung@berkeley.edu

## 1 INTRODUCTION

Supervised learning has proven extremely effective for many problems where large amounts of labeled training data are available. There is a common hope that unsupervised learning will prove similarly powerful in situations where labels are expensive, impractical to collect, or where the prediction target is unknown during training. Unsupervised learning however has yet to fulfill this promise. One explanation for this failure is that unsupervised learning algorithms are typically mismatched to the target task. Ideally, learned representations should linearly expose high level attributes of data (e.g. object identity) and perform well in semi-supervised settings. Many current unsupervised objectives, however, optimize for objectives such as log-likelihood of a generative model or reconstruction error and produce useful representations only as a side effect.

Unsupervised representation learning seems uniquely suitable for meta-learning, or learning how to learn (Hochreiter et al., 2001; Schmidhuber, 1995). Unlike most tasks where meta-learning is applied, unsupervised representation learning does not define an explicit objective, which makes it impossible to phrase the task as a standard optimization problem. It is possible, however, to directly express a *meta-objective* that captures the quality of representations produced by an unsupervised update rule by performing tasks with the representation, e.g. supervised classification. In this work, we propose to *meta-learn* an unsupervised update rule by meta-training on a meta-objective that directly optimizes the utility of the unsupervised representation. Unlike hand-designed unsupervised learning algorithms, this meta-objective directly targets the usefulness of a representation generated from unlabeled data for later supervised tasks.

By recasting unsupervised learning as meta-learning, we treat the creation of the unsupervised update rule as a transfer learning problem. Instead of learning transferable features, such as done in (Vinyals et al., 2016; Ravi & Larochelle, 2016; Snell et al., 2017), we learn a transferable learning rule which does not require access to labels and generalizes across domains. Although we focus on the meta-objective of semi-supervised classification here, in principle a learning rule could be optimized to generate representations for any subsequent task.

## 2 METHOD

We consider a multilayer perceptron (MLP) $f(\cdot; \phi_t)$, with parameters $\phi_t$, as the *base model*. The inner loop of our meta-learning process trains this base model via iterative application of our learned optimizer.

In standard supervised learning, that 'learned' optimizer is stochastic gradient descent (SGD). A supervised loss $l(x, y)$ is associated with this model, where $x$ is a minibatch of inputs, and $y$ are the corresponding labels. The parameters $\phi_t$ are then updated iteratively until convergence by performing SGD using the gradient $\frac{\partial l(x,y)}{\partial \phi_t}$. This supervised update rule can be written as:

$$\phi_{t+1} = \text{SupervisedUpdate}(\phi_t, x_t, y_t; \theta), \tag{1}$$

where $\theta$ are the fixed parameters of the optimizer (e.g. learning rate), which we will refer to as the meta-parameters (also commonly called hyper-parameters).

In this work, our learned optimizer is a parametric update process, which does not depend on label information,

$$\phi_{t+1} = \text{UnsupervisedUpdate}(\phi_t, x_t; \theta). \tag{2}$$

In traditional unsupervised learning algorithms, expert knowledge or a simple hyper-parameter search determines $\theta$, which consists of a handful of meta-parameters such as learning rate, layer sizes, and regularization constants. In contrast, our update rule has many orders of magnitude more meta-parameters such as the weights of a neural network. We train these meta-parameters by performing SGD on meta-objective in order to find optimal parameters $\theta^*$,

$$\theta^* = \operatorname*{argmin}_{\boldsymbol{\theta}} \mathbb{E}_{\text{task}} \left[ \sum_t \text{MetaObjective}(\phi_t(\theta)) \right], \tag{3}$$

that minimize the meta-objective over a set of training tasks.

In this work, our base model consists of fully connected multi-layer perceptron. Our meta objective consists of fitting a linear regression in closed form to one batch of data, and then evaluating on a separate batch. For the learned UnsupervisedUpdate we use a neuron local NN that proposes changes of weights. By structuring our network as neuron local, we are able to apply this on networks of different widths and depths.

Given the size of this extended abstract, we refer a reader to Appendix A for more information on these pieces.

## 3 EXPERIMENTS

We rain an instance of this algorithm on 512 multi-core CPU workers over the course of 5 days with the Adam (Kingma & Ba, 2014) optimizer for 150k steps of meta-training, updates to $\theta$. We approximate $\frac{\partial[\text{MetaObjective}]}{\partial \theta}$ with truncated back-prop for stability as well as compute efficiency. While training, we sample network architectures as well as datasets. As our goal for this work is to learn a transferable algorithm, we shot performance on a variety of generalization tasks.

### 3.1 GENERALIZATION OVER DATASETS AND DOMAINS

The first quantity we wish to generalize over is data set. In Figure 1, we compare performance on few shot classification with 10 examples per class, using embeddings generated by our learned model, by a variational autoencoder with a normal posterior distribution over pixels, as well as by supervised learning. Our model was meta-trained on a distribution of image datasets consisting of downsampled Glyph, Imagenet Russakovsky et al. (2015), and CIFAR10 Krizhevsky & Hinton (2009). We evaluate test performance on higher resolution versions of these datasets, as well as on holdout datasets of MNIST and Fashion MNIST Xiao et al. (2017). Our learned model achieves performance well above random initialization (with learned readout layer), better than supervised learning, and on par or better than a variational autoencoder.

Additionally, we test our learned optimizer on data from a vastly different domain. We train on a binary text classification dataset: IMDB movie reviews (Maas et al., 2011), encoded by computing a bag of words with 1k words. We selected a checkpoint earlier in meta-training, as over-fitting to the image domain occurs later in training. Despite being trained exclusively on image datasets, our learned optimizer improves upon the random initialization by almost 10%. After more unrolling, however, the performance decreases again suggesting that our optimizer has "over-fit" to the image domain. This performance is quite low in an absolute sense. Nevertheless, we find this result very exciting as we are unaware of any work showing this kind of transfer from images to text.

### 3.2 GENERALIZATION OVER NETWORK ARCHITECTURES

To test generalization over neural network architecture we train models of varying depths and unit counts with our learned optimizer and compare results at different points in time. Results can be found in Figure 2. We find that despite only training on networks with 2 to 5 layers and 64 to 512 units per layer, the learned rule generalizes to 11 layers and 10,000 units per layer.

### 3.3 HOW IT LEARNS AND HOW IT LEARNS TO LEARN

To analyze how our learned optimizer functions, we analyze the first layer filters over the course of meta-training. Despite the permutation invariant nature of our data (enforced by shuffling input image

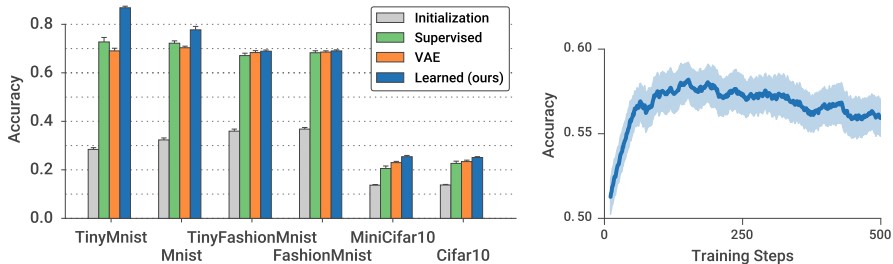

Figure 1: **Left:** The learned optimizer generalizes to unseen datasets (left four bar groups). We additionally show a training dataset, Mini CIFAR10, and a 32x32 sized version (CIFAR10) which has not been seen during training. Our semi-supervised performance is better than both fully supervised learning on the same labeled examples, and a base model representation. Our learned model achieves performance well above random initialization, better than supervised learning, and on par or better than a variational autoencoder. Error bars show standard error. **Right:** Our learned optimizer is able to learn useful features on a 2 way text classification data set, IMDB, despite being trained only from image datasets. Later in training performance drops due to the domain mismatch. Error bars show standard error across 10 runs.

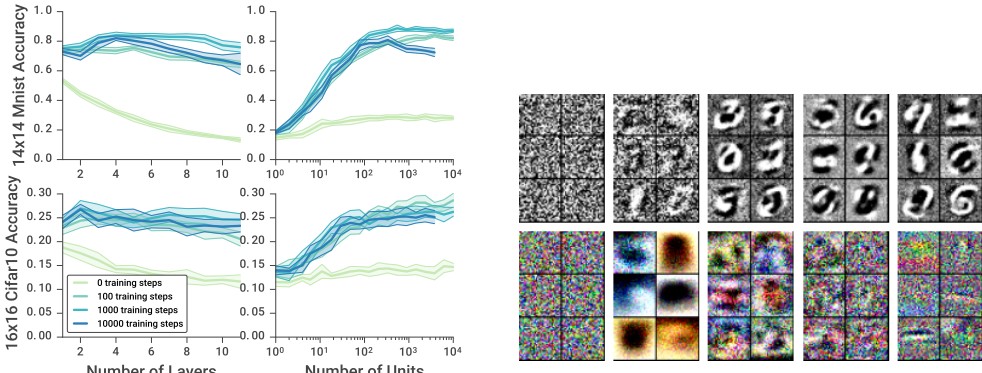

Figure 2: **Left:** The learned optimizer is capable of optimizing base models with hidden sizes and depths outside the meta-training regime. As we increase the number of units per layer, the learned model can make use of this additional capacity despite never having experienced it during meta-training. **Right:** From left to right we show first layer base-model $\phi$ produced by our learned optimizer over the course of meta-training. Each pane consists of first layer filters extracted from $\phi$ after 10k applications of the learned update rule on MNIST (top) and CIFAR10 (bottom). For MNIST, the optimizer learns image template like features. For CIFAR, low frequency features evolve into high frequencies and local edge detectors.

pixels before each unsupervised training run), the base model learns features such as those shown in 2, which appear template-like for MNIST, and local-feature-like for CIFAR10. Early in training, there are course features, and a lot of noise. As the meta-training progresses, more interesting and local features emerge.

ACKNOWLEDGMENTS

We would like to thank Samy Bengio, David Dohan, Gamaleldin Elsayed, Daniel Freeman, Nando de Freitas, Ross Goroshin, Ishaan Gulrajani, Eric Jang, Hugo Larochelle, Esteban Real, Pavel Sountsov, Alex Toshev, George Tucker, Olga Wichrowska, and Lechao Xiao for extremely helpful conversations and feedback on this work.

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

## A  Methods More Detail

### A.1  Base Model: $f(\cdot; \phi)$

Our base model consists of a standard fully connected multi-layer perceptron (MLP), with batch normalization (Ioffe & Szegedy, 2015), and ReLU nonlinearities. We call the pre-nonlinearity activations $z^1..z^L$, and post-nonlinearity activations $x^0..x^L$, where $L$ is the total number of layers, and $x^0 \equiv x$ is the network input (raw data). $W^l$ and $b^l$ are the weights and biases for layer $l$.

### A.2  Learned update rule: SupervisedUpdate$(\cdot; \theta)$

We wish for our update rule to generalize across architectures with different widths, depths, or even network topologies. To achieve this, we design our update rule to be *neuron-local*, so that updates are a function of pre- and post- synaptic neurons in the base model, and are defined for any base model architecture. This has the added benefit that it makes the weight updates more similar to synaptic updates in biological neurons, which depend almost exclusively on the pre- and post-synaptic neurons for each synapse (Whittington & Bogacz, 2017).

To build these updates, each neuron $i$ in every layer $l$ in the base model has an MLP, referred to as an update network, associated with it, with output $h_i^l(\cdot; \theta)$. All update networks share meta-parameters $\theta$, and $h_i^l(\cdot; \theta)$ is evaluated only during unsupervised training as the update networks are part of the SupervisedUpdate, and not part of the base model. Evaluating the statistics of unit activation over a batch of data has proven helpful in supervised learning (Ioffe & Szegedy, 2015). It has similarly proven helpful in hand-designed unsupervised learning rules, such as sparse coding and clustering. We therefore allow $h_i^l(\cdot; \theta)$ to accumulate statistics across examples in each training minibatch.

During an unsupervised training step, the base model is first run in a standard feed-forward fashion, populating $x_{ib}^l, z_{ib}^l$, where $b$ is the training minibatch index. As in supervised learning, an error signal $\delta_{ib}^l$ is then propagated backwards through the network. Unlike in supervised backprop however, this error signal is generated by the corresponding update network for each unit, $\delta_{ib}^l \leftarrow h_i^l(\cdot; \theta)$. Again, as in supervised learning, the weight updates are a product of pre- and post-synaptic signals. Unlike in supervised learning however, these signals are also generated from the update networks: $\Delta W_{ij}^l = \sum_b c_{ib}^l d_{jb}^{l-1}$, where $\{c_{ib}^l, d_{ib}^l\} \leftarrow h_i^l(\cdot; \theta)$. The inputs to the update network consists of unit pre- and post-activations, and backwards propagated error signal: $h_i^l\left(x_{i\cdot}^l, z_{i\cdot}^l, \left[\left(W^{l+1}\right)^T \delta^{l+1}\right]_{i\cdot}; \theta\right)$.

In practice, we also include lateral interaction terms to aid units within the same layer in remaining decorrelated from each other. These enter as additional contributions to $\Delta W_{ij}^l$, and as additional inputs to $h_i^l(\cdot; \theta)$ not described in this short submission.

### A.3  MetaObjective $(\phi)$

The meta-objective determines the quality of the unsupervised representations. In order to meta-optimize with SGD, the evaluation of this loss must be differentiable. The meta-objective we use in this work is based on fitting a linear regression to labeled examples with a small number of data points. In order to encourage the learning of features, which generalize well, we estimate the linear regression weights on one minibatch $\{x_a, y_a\}$ of $K$ data points, and evaluate the classification performance on a second minibatch $\{x_b, y_b\}$ also with $K$ datapoints,

$$\hat{v} = \underset{v}{\operatorname{argmin}} \left(\left\|y_a - v^T u_a\right\|^2 + \lambda \left\|v\right\|^2\right) \tag{4}$$

$$\text{MetaObjective}(\cdot; \phi) = \text{CosDist}\left(y_b, \hat{v}^T u_b\right), \tag{5}$$

where $u_a$, $u_b$ are features extracted from the base model on data $x_a$, $x_b$, respectively. The target labels $y_a$, $y_b$ consist of one hot encoded labels and potentially also regression targets from data augmentation (e.g. rotation angle). We found that using CosDist rather than unnormalized squared error dramatically improves stability.

