# OpenReview forum: "Learning to Learn Without Labels"
_ICLR.cc/2018/Workshop — Accept_

### Official Review · AnonReviewer2 · 2018-03-10
**Well-written paper, good to accept.**

**Rating:** 7
**Confidence:** 3

**Review:**

This paper introduces an unsupervised meta learning approach by learning the update rules in the unsupervised phase and transfer them to the target tasks. The framework consists of 3 elements: a meta objective, a base model and a set of learned update rules, each of which is also a neural network. In the unsupervised learning phase, the set of update rules is trained to optimize the base model by minimizing the meta objective. Then, the learned update rules are used to optimize new models during supervised learning.

In the experiments, the update rules and the base model are trained on a variant of Imagenet and CIFAR 10. Then, the base model is used to generate representation to perform low shot learning on various datasets. The results show that classifiers trained on such embedding are consistently better than random initialization and slightly better than VAE initialization.

In addition, the set of update rules are applied to optimize networks with various architectures. The results show that the update rules can optimize networks that are bigger and deeper than the base model and exploiting the additional capacity although they are only trained to optimize the base model during the meta training phase.

The paper also shows that their method can transfer knowledge from image to text domain, but the accuracy tends to drop with more training iteration due to domain mismatch.  Authors can conduct more extensive analysis, such as why the base model can generalize from image to text, what are the common features that the base model has learned to accomplish this, and how to avoid domain mismatch. Finally, it will be important to explain the MetaObjective in A.3. It is a bit unclear how to obtain the target label y given it is an unsupervised learning approach.

Overall, this is a well-written paper with interesting idea and positive results for acceptance.

---

### Official Review · AnonReviewer3 · 2018-03-10
**Promising and original idea**

**Rating:** 8
**Confidence:** 3

**Review:**

This work presents a new approach to unsupervised representation learning with neural nets. The unsupervised learning problem is seen as a meta-learning problem where an update rule for the base model is learnt through a supervised meta-objective (e.g. linear regression fit on mini-batches in the experiments). Results obtained on few-shot classification shows competitive or better performance compared to the VAEs based approach. The proposed approach appears promising and original.

Questions and comments:

The paper is dense and relatively difficult to follow, though it might be hard to do better with the space constraints.

In equation (3), why consider the sum over time steps instead of the last step? It would be good to spell out what theta exactly in this case.

It would also be good to define what neutron local means.

---

### Official Review · AnonReviewer1 · 2018-03-11
**meta-learning without label (good idea to explore)**

**Rating:** 6
**Confidence:** 4

**Review:**

This paper proposed a meta-learning method with unlabeled data. The paper argues that unsupervised learning has been suffering from not properly defined objective. This paper proposes to combine Mata-learning with unsupervised learning to make sure that the learned representation is meaningful.
I enjoy this idea. However, this work is still in its early stage.

Detailed Comments:

1. It is maybe a bit over claiming to say "without labels" since the method does need few data with labels for the linear regression.
2. In the paper, the meta objective is only applied among two mini-batches of data. When it comes to many mini batches, how does it work? Will it encounter some side-effects?
3. I am not sure whether the experiments are fair. Meta-learning uses more data than supervised learning, right? They just share the same amount of labeled data.
4. Please explain more about the "over-fit" in the second part of 3.1. This is a rather challenging task. How does the method over-fit?
5. Typos in the paper, such as "rain"-> "train"

---

### Decision · Program_Chairs · 2018-03-20
**ICLR 2018 Workshop Acceptance Decision**

**Decision:**

Accept

**Comment:**

Congratulations, your paper was accepted to the ICLR workshop.